# Antisite occupation induced single anionic redox chemistry and structural stabilization of layered sodium chromium sulfide

Zulipiya Shadike[1,2], Yong-Ning Zhou[3], Lan-Li Chen[4], Qu Wu[4], Ji-Li Yue[1], Nian Zhang[5], Xiao-Qing Yang[3], Lin Gu[6], Xiao-Song Liu[5], Si-Qi Shi[4,7] & Zheng-Wen Fu[1]

The intercalation compounds with various electrochemically active or inactive elements in the layered structure have been the subject of increasing interest due to their high capacities, good reversibility, simple structures, and ease of synthesis. However, their reversible intercalation/deintercalation redox chemistries in previous compounds involve a single cationic redox reaction or a cumulative cationic and anionic redox reaction. Here we report an anionic redox chemistry and structural stabilization of layered sodium chromium sulfide. It was discovered that the sulfur in sodium chromium sulfide is electrochemically active, undergoing oxidation/reduction rather than chromium. Significantly, sodium ions can successfully move out and into without changing its lattice parameter $c$, which is explained in terms of the occurrence of chromium/sodium vacancy antisite during desodiation and sodiation processes. Our present work not only enriches the electrochemistry of layered intercalation compounds, but also extends the scope of investigation on high-capacity electrodes.

[1] Shanghai Key Laboratory of Molecular Catalysts and Innovative Materials, Department of Chemistry & Laser Chemistry Institute, Fudan University, Shanghai 200433, China. [2] Chemistry Division, Brookhaven National Laboratory, Upton, NY 11973, USA. [3] Department of Material Science, Fudan University, Shanghai 200433, China. [4] School of Materials Science and Engineering, Shanghai University, Shanghai 200444, China. [5] State Key Laboratory of Functional Materials for Informatics, Shanghai Institute of Microsystem and Information Technology, Chinese Academy of Science, Shanghai 200050, China. [6] Beijing National Laboratory for Condensed Matter Physics, Institute of Physics, Chinese Academy of Sciences, Beijing 100190, China. [7] Materials Genome Institute, Shanghai University, Shanghai 200444, China. Zulipiya Shadike, Yong-Ning Zhou and Lan-Li Chen contributed equally to this work. Correspondence and requests for materials should be addressed to X.-S.L. (email: xliu3@mail.sim.ac.cn) or to S.-Q.S. (email: sqshi@shu.edu.cn) or to Z.-W.F. (email: zwfu@fudan.edu.cn)

 1

Since the first commercialization in 1991, rechargeable lithium-ion battery (LIB) has powered most consumer electronic devices because of their high gravimetric and volumetric energy densities. LIB has also emerged as a key technology for electric vehicles and has been considered as a good candidate for grid-scale stationary energy storage. Over the past decade, designing and optimizing intercalation cathode materials including the layered oxides ($LiMO_2$, $NaMO_2$), spinel oxides ($LiM_2O_4$), and olivine phosphates $LiMPO_4$ (M = transition metals) have contributed greatly in developing new electrode materials for high performance secondary batteries[1–3]. The fundamental strategy for designing intercalation cathodes used in secondary battery is on the basis of the reversible deintercalation/ intercalation of guest ions from/into the host framework without changing the skeleton structure. In general, most of layered intercalation compounds for secondary batteries involve the cationic reversible redox processes[4–7]. These compounds, like $LiCoO_2$[4], $NaCrO_2$[5], $LiNi_{1/3}Mn_{1/3}Co_{1/3}O_2$[6], and $LiMS_2$[7] (M = Ti, V, Cr), can be described as an alternate layer structure with alkali cation sheets sandwiched between transition-metal oxide/sulfide slabs. The transition metal ions as the redox centers could be reduced to a lower oxidation state by electrochemical intercalation of guest ions. Recently, the contributions from anionic redox reactions are reported in high-capacity layered oxides and gaining increasing attentions[8–12]. Some Li-rich cathode materials, such as $Li_{1.20}Mn_{0.54}Co_{0.13}Ni_{0.13}O_2$[8], $Li_2Ru_{1-y}Sn_yO_3$ $(0 < y < 1)$[9], $Li_2Ru_{1-y}Ti_yO_3$ $(0 \leq y \leq 1)$[10], $Li_4FeSbO_6$[11], and $Li_2Ir_{1-x}Sn_xO_3$ $(0 < x < 1)$[12], exhibit a cumulative cationic and anionic ($O^{2-}/O_2^{2-}$) reversible redox chemistry when Li ions reversibly deintercalate/intercalate from/into these compounds. From both experimental[8] and theoretical[13] points of view, recent studies revealed that the anionic redox was triggered by forming

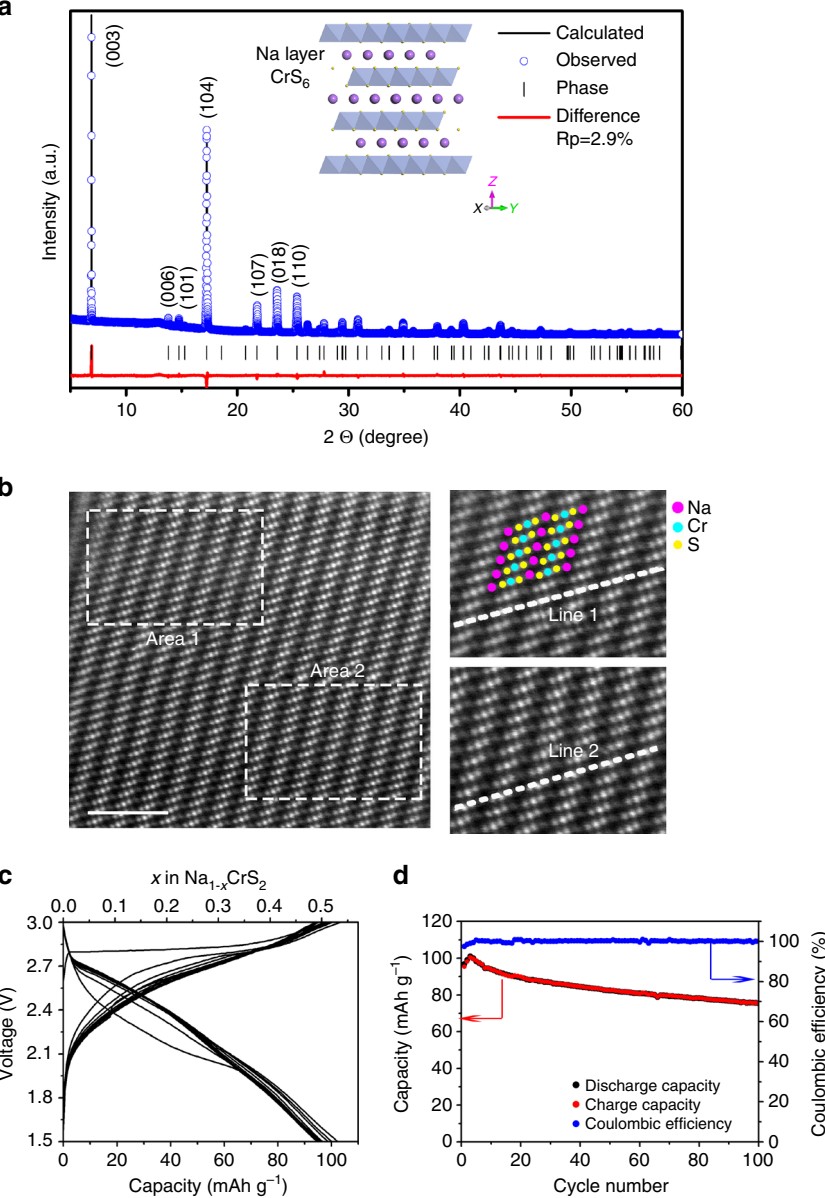

**Fig. 1** Structural, morphological, and electrochemical performance of $NaCrS_2$. **a** XRD patterns of powder $NaCrS_2$ (*blue circle*), calculated profile (*black solid line*), and their difference (*red solid line*). Bragg positions are indicated as *black vertical tick marks*. In the inset for the structure schematic of $NaCrS_2$, legend: *purple* (Na), *blue* (Cr), and *yellow balls* (S); **b** HAADF-STEM image of the pristine $NaCrS_2$ particle, *scale bar* 2 nm; **c** Galvanostatic charge/discharge curves, **d** cyclic performances and Coulombic efficiencies of $NaCrS_2$ electrode at 0.5 C

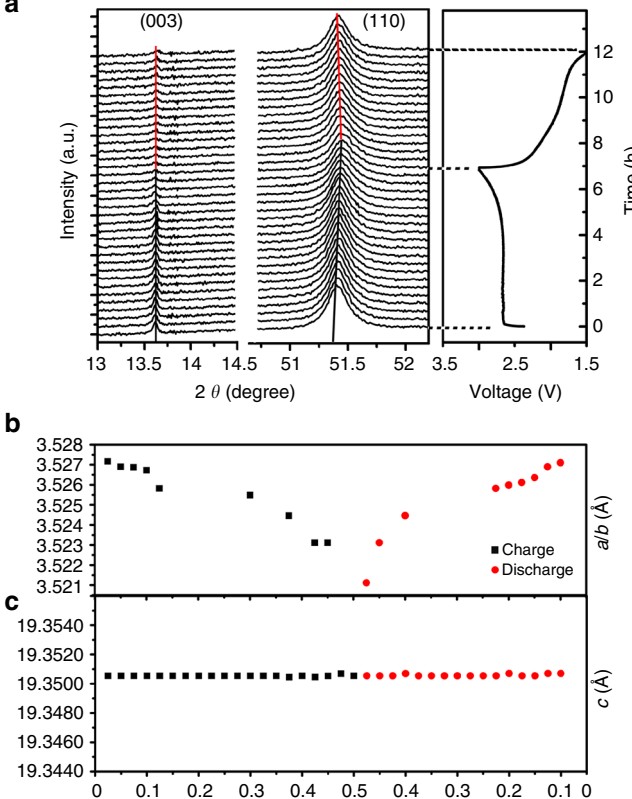

**Fig. 2** Structural evolution of NaCrS$_2$ during sodium deintercalation and intercalation. **a** In situ XRD during the initial cycle for NaCrS$_{2,}$ corresponding voltage curve is shown on the *right*. The 2θ is converted to the corresponding angle for λ = 1.54 Å (Cu-Kα) from the real wavelength λ = 0.7747 Å used for synchrotron XRD experiments. Evolution of lattice parameters **b** a/b and **c** c calculated from the in situ XRD during the first charge/discharge process

non-bonding oxygen states, which was created due to that those O 2p orbitals in Li-O-Li configuration have no transition metal orbitals with which it can hybridize. The oxidation of oxygen takes place on the orphaned electrons in Li-O-Li configuration. The increased capacity arises from the fact that two redox bands can be accessed by MO* states and non-bonding oxygen states in Li-rich layer-structured and cation-disordered cathode materials[14]. Although this so-called "anion-cation redox competition" behavior was first discussed by Rouxel in 1996[15] with some less-electronegative chalcogenides, the current strategies for designing new high-capacity intercalation-type electrodes are mainly limited to the optimized transition metal redox properties and the utilization of the redox of oxygen. Recently, Du and Goodenough et al.[16] investigated the sole anionic redox in a P3-layered Na$_{0.6}$(Li$_{0.2}$Mn$_{0.8}$)O$_2$, and found that the holes in O-2p bands were introduced during desodiation, but cannot cycle reversibly. Here, a sole reversible anionic redox chemistry of S$^{2-}$/S$_2^{2-}$ is triggered by the Cr/V′$_{Na}$ (Na vacancy: V′$_{Na}$ in Kröger-Vink notation) antisite, without the redox of transition metal. The relationship between the redox chemistry of S$^{2-}$/S$_2^{2-}$ and its electronic structure is systematically investigated with multi-probe experimental characterizations and theoretical calculations. The motivation of this work is to enrich the reversible redox chemistry of layer-structured intercalation compounds for understanding the nature of their structural and electronic property variations during the deintercalation/intercalation of guest ions.

In this work, the deintercalation/intercalation reaction of NaCrS$_2$ as a model-layered compound with the typical S$^{2-}$/S$_2^{2-}$ redox is presented. Typical reversible redox chemistry of sulfur, an "abnormal" unit cell breathing behavior and Cr/V′$_{Na}$ antisite are observed during the first desodiation/sodiation processes. Density functional theory (DFT) calculations reveal that the top of valance band of NaCrS$_2$ is mainly populated by S electrons. Therefore, sulfur undergoes redox chemistry for charge compensation during sodium removal. Besides, the occurrence of Cr/V′$_{Na}$ antisite triggers the redox of sulfur and is found to be responsible for the "abnormal" unit cell breathing without changing its lattice parameter c.

## Results

**Structural, morphological, and electrochemical performance.** The crystal structure of NaCrS$_2$ is measured by synchrotron X-ray diffraction (XRD). The refined pattern is presented in Fig. 1a. All Bragg reflections are indexed by using a rhombohedral symmetry with space group *R-3m* (No. 166). This is a typical O3-type layered structure (a = b = 3.5270(4) Å, c = 19.3506(1) Å) and no crystalline impurities are observed. The crystal structure schematic is illustrated in *inset* of Fig. 1a. The Rietveld refinement shows reasonable small R factors (R$_p$ = 2.9%, R$_{wp}$ = 3.9%) and high goodness of fit (GOF (χ$^2$) = 4.67). The structure parameters are listed in Supplementary Table 1. The high intensity ratio (1.55) of (003)/(104) and clearly separated (110) and (018) peaks imply a clear layered O3 structural configuration without Cr/Na antisite defects between the oxygen layers[17–19]. It is also confirmed by TEM observations at atomic-scale resolution on the pristine NaCrS$_2$ (Fig. 1b and Supplementary Figs. 1 and 2). Figure 1b exhibits the typical image of high-angle-annular-dark-field (HAADF) for NaCrS$_2$ sample at atomic-level projected along [100] direction. Line scans are performed on the surface (line 1) and bulk (line 2) of a particle. The corresponding contrast curves from the line scans are presented in Supplementary Fig. 1a, b. The peaks in the curves correspond to Cr, S, and Na atomic columns in the HAADF image. The intensity ratio between Na and Cr column is ~14.59%. From STEM images and contrast curves of line scans, it can be concluded that the structural distribution is homogeneous from surface to bulk of the NaCrS$_2$ particles. NaCrS$_2$ particles are in irregular shape with a size range of 2–4 μm (Supplementary Fig. 3).

Figure 1c shows the charge and discharge curves of NaCrS$_2$ at 0.5 C. During the initial charge, NaCrS$_2$ electrode displays a flat voltage at 2.85 V and delivers a high capacity of 95 mAh g$^{-1}$ (0.49 Na$^+$ per NaCrS$_2$), which is smaller than that of Na$_x$VS$_2$ and Na$_x$TiS$_2$ electrodes[20] (160 mAh g$^{-1}$). The initial discharge profile includes a slopy region from 2.8 to 2.1 V and a flat plateau at 2.0 V. The first discharge capacity 92 mAh g$^{-1}$ is presented. In subsequent three cycles, the charge and discharge capacities gradually increase due to the activation process of electrode, e.g., the specific charge and discharge capacity of 104.1 and 103.4 mAh g$^{-1}$ is obtained, respectively, in the third cycle. The coulombic efficiency is 99.3%. The cyclic performance and coulombic efficiency of NaCrS$_2$ electrode are presented in Fig. 1d. The capacity keeps decreasing after the 50 cycles. A capacity of 78.5 mAh g$^{-1}$ can still be obtained after 100 cycles, whereas the coulombic efficiency increases during the initial 10 cycles and then reaches almost 100% after 100 cycles. At 0.1 C shown in Supplementary Fig. 4, the initial charge capacity is 93.2 mAh g$^{-1}$. The reversible capacity also increases in a few cycles at the beginning and gets to the maximum value 107.3 mA h g$^{-1}$, then decreases to 78.6 mAh g$^{-1}$ at the 100th cycle. The first five cyclic voltammograms of NaCrS$_2$ electrode are presented in Supplementary Fig. 5. It shows that the cathodic/anodic

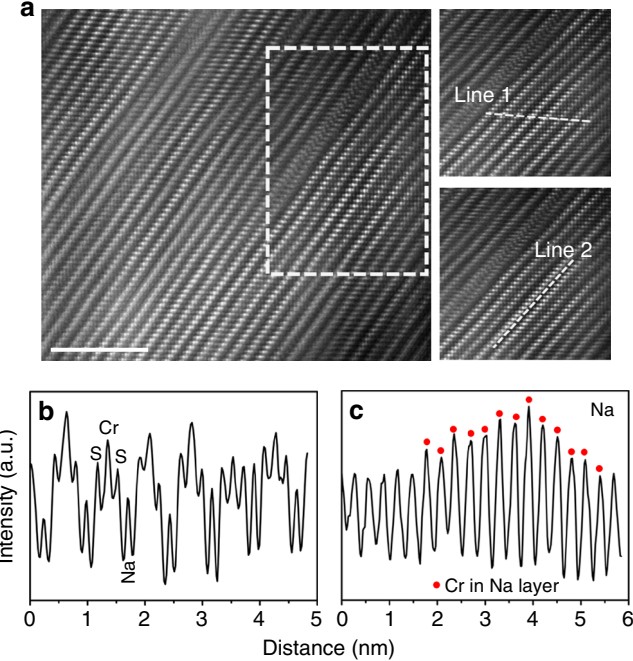

**Fig. 3** Atomic structure of $NaCrS_2$ after charge. **a** HAADF-STEM image of the desodiated particle ($Na_{1-x}CrS_2$: $x = 0.5$ calculated from capacity), *scale bar* 5 nm. **b, c** The contrast profiles of the regions scanned by *line 1* (Cr, S, and Na ions columns) and *2* (Na columns), respectively. (*Red circle* represents Cr ion in the Na layer)

peaks emerge at 2.85/2.05 V in the first cycle, which are close to the galvanostatic charge/discharge voltage plateaus in the first cycle in Fig. 1c.

**Na-driven structural evolution during cycling**. To monitor the structural changes of $NaCrS_2$ electrode upon sodium deintercalation/intercalation, synchrotron-based in situ XRD is carried out. Figure 2a shows a series of XRD patterns obtained in the initial cycle. In this work, the fully charged state of the $Na_{1-x}CrS_2$ electrode is defined by the samples charged to 3.0 V ($Na_{0.5}CrS_2$). According to the crystallography theory, (003) and (110) peaks reflect changes along $c$ and $a$ axes of the layered structure, respectively. During the charge process, the (110) peak gradually moves toward higher $2\theta$ angles when $x$ value is increased from 0.0 to 0.5 in $Na_{1-x}CrS_2$, indicating a solid solution reaction with the continuous lattice contraction along the $a$ axis. During the discharge process, the (110) peak reversibly moves back to its original position when $x$ value decreases from 0.5 to 0 in $Na_{1-x}CrS_2$. Very interestingly, the $2\theta$ angle of (003) diffraction peak keeps unchanged in the entire charge/discharge processes, implying that the lattice parameter $c$ remains almost constant. The lattice parameter evolution of $NaCrS_2$ electrode during the initial cycle is shown in Fig. 2b, c. From $x = 0$ to 0.5 during charge process, the lattice parameter $a$ decreases from 3.527 to 3.521 Å and then reversibly increases back to 3.527 Å during the discharge process. On the contrary, the lattice parameter $c$ keeps unchanged, resulting in a very small unit cell volume change during desodiated/sodiated process. The structure evolution and lattice parameter change of $NaCrS_2$ electrode during the third cycle are found to be almost consistent with those during the first cycle (Supplementary Fig. 6). No other peak is observed, indicating that no new structure is formed in the subsequent cycles. The invariableness of lattice parameter $c$ of $NaCrS_2$ electrode during charge/discharge process is quite unusual compared with many

other layered cathode materials, in which the parameter $c$ is mostly increasing during charge and decreasing during discharge known as "normal" unit cell breathing behavior[21]. To our knowledge, the unchanged parameter $c$ was only observed in layered $LiNbO_2$ system[22]. In the layered cathodes with "normal" unit cell breathing behavior, the increasing lattice parameter $c$ is due to the expansion of the interlayer spacing caused by the enhanced repulsion force between the two neighboring oxygen layers in delithiation (or desodiation). However, in the $LiNbO_2$ system, the expansion of the interlayer spacing is compensated by the contraction of the $NbO_6$ trigonal prism along the $c$ axis because of the deformation of $NbO_6$ trigonal prism during charge. Because the layered structure of $NaCrS_2$ with edge-sharing of $CrO_6$ octahedra is different from that of $LiNbO_2$ with edge-sharing $NbO_6$ trigonal prisms, the mechanism of unchanged parameter $c$ in layered $LiNbO_2$ could not be applied for $NaCrS_2$.

In order to understand the "abnormal" unit cell breathing mechanism of $NaCrS_2$ electrode during the first charge process, the structure of the first charged sample at atomic scale is investigated by STEM. As shown in Fig. 3a, the contrast of Na, Cr, and S atomic columns changes a lot, compared with the pristine sample. The line profiles of Fig. 3b, c are associated with the dash line 1 and 2, respectively. The intensity ratio between Na and Cr column is ~49.81%, which is higher than the ratio in pristine $NaCrS_2$ sample, implying the existence of cation mismatch. Besides, the intensity fluctuation of Na column confirms the uneven Cr occupation in $Na_{0.5}CrS_2$ (red cycle represents Cr ion in the Na layer). Similar phenomena have been observed in $LiCrO_2$ and $LiVO_2$ electrodes[23, 24]. Such Cr migration into Na sites may be responsible for the unchanged lattice parameter $c$.

**Detection of Cr valance state**. Ex situ Cr K-edge XAS experiments are carried out to examine valance states of Cr during charge and discharge. The *red blocks* on the *curves* in Fig. 4a mark the positions for XAS spectrum collection. The X-ray absorption near-edge structure (XANES) spectra of Cr K-edge at various charge/discharge states are presented in Fig. 4b. No obvious change of the spectra is observed during the initial cycle, indicating that very limited redox reaction for Cr ions takes place during the entire electrochemical cycle. This phenomenon is quite different from the other layer-structured cathodes[25]. The Fourier transformed extended X-ray absorption fine structure (FT-EXAFS) spectra of the pristine and fully charged electrodes are shown in Fig. 4c, d. The first peak at ~2.0 Å is attributed to the single scattering path from the closest S ions to the core Cr ions, and the peak at ~3.25 Å is due to the scattering from the nearest Cr ions in Cr layer. The $R$ values of these peaks are about 0.3–0.4 Å shorter than the real bond lengths, because they are not phase corrected[25]. Supplementary Table 2 lists the structure parameters derived from fitting. It is observed that the atomic distances of both Cr-S and Cr-Cr are decreasing upon charge, implying the shrinking of $Cr-S_6$ octahedrons and $Cr-Cr_6$ hexagons. These are consistent with XRD results showing that the lattice parameter $a$ decreases during the charge process.

**EPR spectra**. EPR spectroscopy is employed to further monitor the sodium content-induced changes in the oxidation states of Cr or S ions in $Na_xCrS_2$ series, and Fig. 4f shows the EPR spectra of $Na_xCrS_2$ series at room temperature (RT). The $Cr^{3+}$ ion in $NaCrS_2$ has the $3d^3$ electron configuration, which is EPR-active owing to existence of unpaired electrons, while $S_2^{4-}$ in $NaCrS_2$ is EPR-inactive, because all electrons are paired[26]. All spectra consist of single Lorentzian line, and $g$ factor is calculated using

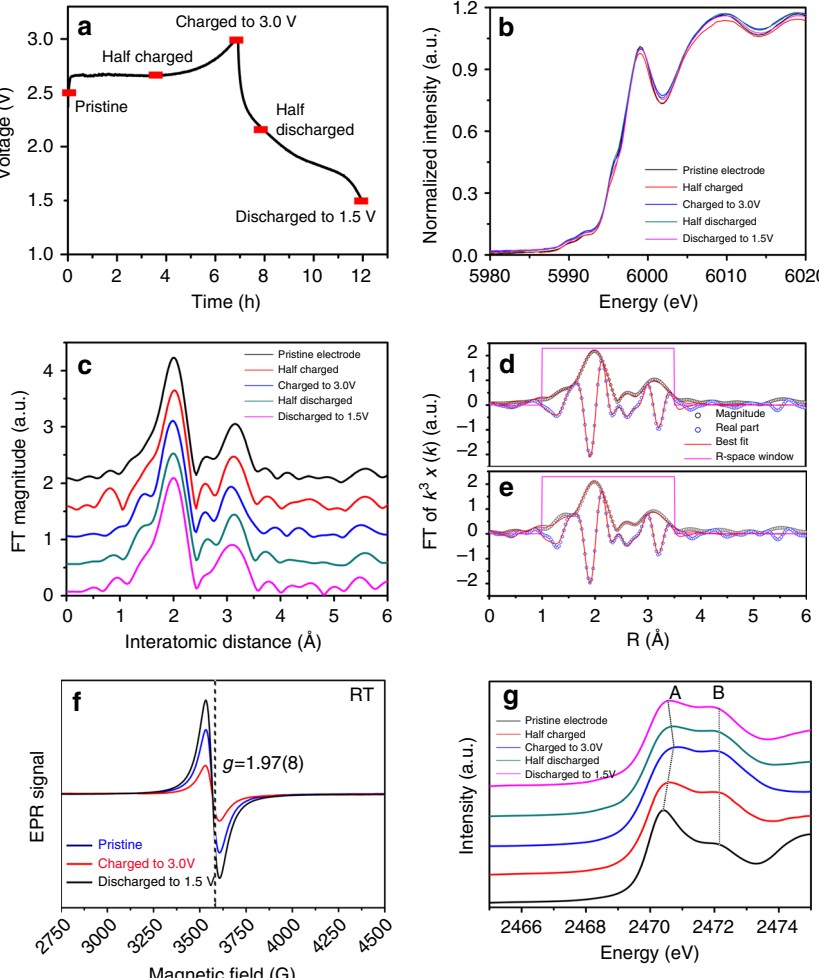

**Fig. 4** Detection of Cr and S valance state during various charge-discharge stages. **a** Charge and discharge curves of NaCrS$_2$ electrode for ex situ XAS. **b** Cr K-edge XANES spectra of NaCrS$_2$ electrodes at various states. **c** Corresponding FT-EXAFS spectra of the NaCrS$_2$ electrode at various charge and discharge stages. Calculated (*solid red line*) and experimental (*solid and open circle*) FT-EXAFS spectra for **d** pristine and **e** fully charged samples. **f** EPR spectra of pristine NaCrS$_2$, fully charged and discharged samples recorded at room temperature. **g** S K-edge XANES spectra of NaCrS$_2$ at the initial cycle

the following relationship[27]:

$$hv_r = g\mu_B H_r. \qquad (1)$$

The effective $g$ value of 1.97(8) in the magnetic field of 3570 G obtained from RT is associated with contributions from both exchange-coupled pairs of Cr$^{3+}$[28, 29]. It is undoubted that the Cr$^{3+}$ EPR-signal and $g$ factor are almost unchanged during charge/discharge processes, indicating the valence state of Cr ions remains unchanged during the first cycle. In addition, no S-signal is detected in the charged sample, which can be explained by the formation of (S$_2$)$^{2-}$ species that have no unpaired electron as shown in Supplementary Fig. 8, thus are EPR-inactive.

**S K-edge XANES**. To examine valence states of S for NaCrS$_2$ during cycling, ex situ S K-edge XANES spectra are measured. As presented in Fig. 4g, two peaks at around 2470.3 and 2472.1 eV in S K-edge XANES spectra of the pristine NaCrS$_2$ are observed, corresponding to these unoccupied $t_{2g}$ and $e_g$ bands, respectively[30]. These bands are derived from the hybridization of sulfur $3p$ states and delocalized Cr $3d$ states. The peak distance (band splitting of $t_{2g}$-$e_g$) about 1.8 eV is close to that of LiTiS$_2$[31]. In the charge process, the shapes of S K-edge XANES spectra change obviously. A gradually positive shift of the edge peak A to high energy is clearly observed, and there is a large displacement

of about 0.4 eV between the pristine and fully charged states. More interestingly, the intensities of peak B gradually increase during charge. Previous studies reported that a significant increase of the edge peak intensity in S K-edge XANES spectra for lithium deintercalation in Li$_x$TiS$_2$ ($0 \leq x \leq 1$) is indicative of the electron transfer involving in sulfur[31, 32]. The position shifts of peak A and the intensity increments of peak B during charge should mainly result from the contributions of the formation of new chemical bonds to the sulfur atoms. This new peak might emerge around 2742.1 eV and its intensity increases during the charge process. Therefore, such a new peak indicates the oxidation of S$^{2-}$ during the sodium deintercalation process. These variations in S K-edge XANES spectra show reversibility during discharge process, which strongly confirm that the spectral evolution is related to the electrochemical sodium intercalation/deintercalation reactions. From the results above, it can be deduced that charge compensation of NaCrS$_2$ during charge and discharge is mainly achieved by the redox of S$^{2-}$.

**S $2p$ XPS spectra**. To further confirm the electron structure of S during charge and discharge, XPS measurements are carried out on NaCrS$_2$ at pristine (Fig. 5a), fully charged (Fig. 5b), and fully discharged states (Fig. 5c). For the pristine sample, the S $2p$ spectra can be deconvoluted into three peaks: two peaks at

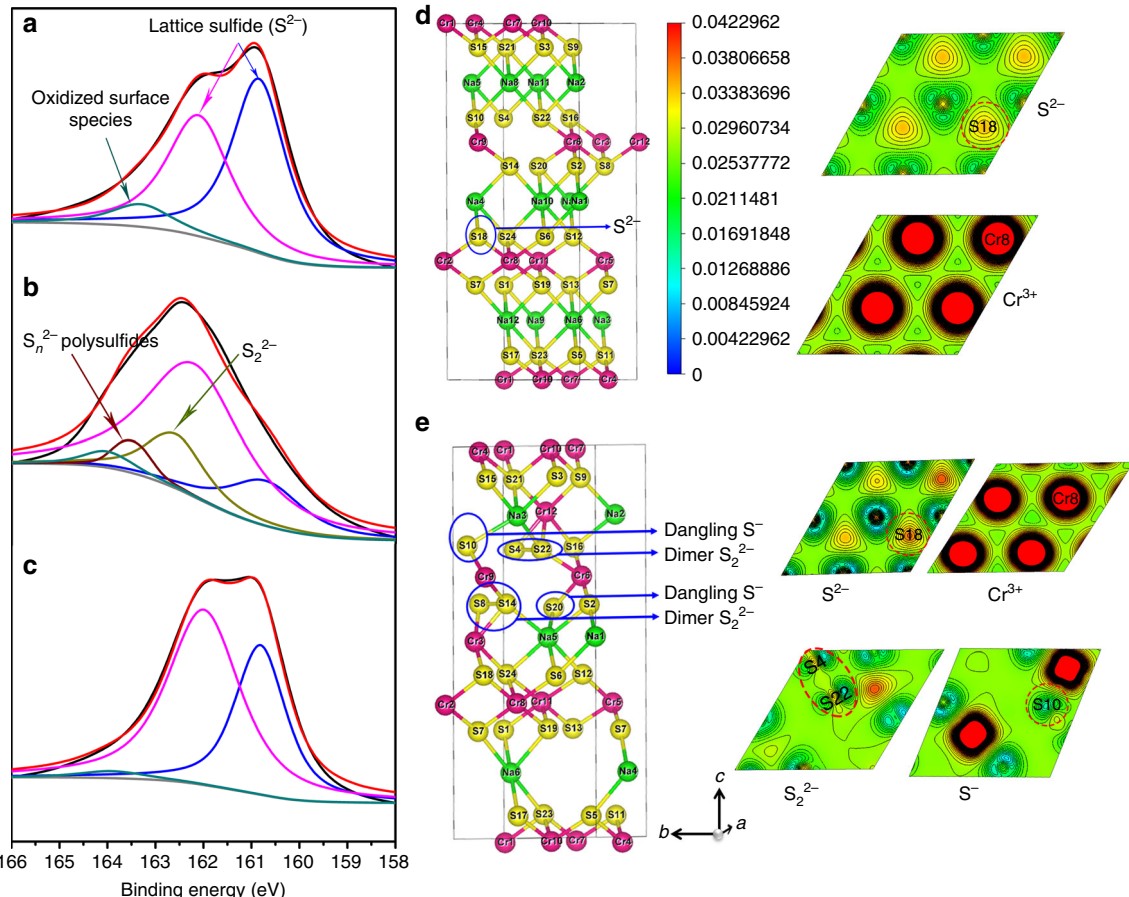

**Fig. 5** Detection of S species evolution during various charge-discharge stages. S $2p$ XPS spectra of NaCrS$_2$ samples, **a** pristine, **b** after charging to 3.0 V, and **c** after discharging to 1.5 V. Crystal structures and electron density contours of **d** pristine NaCrS$_2$ and **e** Na$_{0.5}$CrS$_2$ with 1/6 Cr/V′$_{Na}$ antisite (2 × 2 × 1 conventional cells). Selected planes are through the Cr and S ions labeled in crystal structures. The contour unit is e/Bohr$^3$ (1 Bohr = 0.529 Å)

160.9 and 162.2 eV are attributable to the lattice sulfide S$^{2-}$, while the peak at 164.0 eV can be explained from certain surface oxidization of species[33, 34]. After full charge (Fig. 5b), the majority of S$^{2-}$ is oxidized to higher valance states with higher binding energies as expected. The new peaks at 162.6 and 163.6 eV are well assigned to S$_2^{2-}$ and short-chain polysulfides (S$_n^{2-}$, 2 < n < 8)[35–37], respectively. After discharge (Fig. 5c), it is obvious that two peaks from S$_2^{2-}$ disappear along with the spectrum shape recovering to that of the initial, further confirming reversible redox of sulfur during charge and discharge processes.

**DFT calculations**. Figure 6a–c shows the density of states of Cr-3$d$ and S-3$p$ in pristine NaCrS$_2$. The band gap of 1.27 eV for NaCrS$_2$ is close to that of 1.16 eV obtained with the atomic sphere approximation[38]. Furthermore, the calculated magnetic moment of pristine NaCrS$_2$ (3.322 $\mu_B$) is slightly larger than the experimental value (3 $\mu_B$)[38]. Figure 6b, c also shows that the vast majority states ranging from −5 to −0.49 eV are occupied by Cr-3$d$ and these states are mainly contributed by $t_{2g}$, which further confirms that the electron configuration of Cr$^{3+}$ is the $t_{2g}^3(\uparrow)$ $e_g^0(\uparrow)$. Most important is that the majority states from −0.49 to the Fermi level ($E_F$) and the minority states from −5 eV to $E_F$ originate from S-3$p$ band (See the *inset* in Fig. 6a). This is in excellent agreement with the angle-resolved photoemission experiment revealing Cr-3$d$ bands below the top of S-3$p$ bands[39].

The ground state configuration of Na$_{1-x}$CrS$_2$ with different Na concentrations ranging from $x = 0$ to $x = 0.5$ are determined by cluster expansion and the results are shown in Supplementary Figs. 9–13. Based on the ground state configuration of Na$_{0.5}$CrS$_2$ (without Cr/V′$_{Na}$ antisite) obtained from the cluster expansion and by referring to the XRD and STEM results in Figs. 2 and 3, we build 103 possible configurations with Cr/V′$_{Na}$ antisite (1/6 Cr in the V′$_{Na}$), whose lattice parameters and total energies are summarized in Fig. 6d–f and Supplementary Tables 3 and 4. As shown in Fig. 6, lattice parameters $a'$ ($a'$: lattice parameter of 2 × 2 × 1 conventional cell, $a'$ is equal to 2$a$) of all configurations are smaller by 0.02376–0.2297 Å than that of pristine NaCrS$_2$. Interestingly, a configuration (*red square*) shows a lowest total energy and the $c$ axis length is close to that of pristine NaCrS$_2$ (19.342 vs. 19.464 Å), but shorter by 1.205 Å than that of Na$_{0.5}$CrS$_2$ without Cr/V′$_{Na}$ antisite. Actually, as compared with a sharp increase of lattice parameter $c$ of Na$_{1-x}$CrS$_2$ without antisite (from 19.464 Å at $x = 0$ to 20.622 Å at $x = 1/2$) as shown in Supplementary Fig. 14, the almost constant lattice parameter $c$ of Na$_{1-x}$CrS$_2$ with 1/12 Cr/V′$_{Na}$ antisite is also obtained. Besides, the lattice parameter changes due to the desodiation for NaCrS$_2$ ($\Delta a = -0.17\%$ and $\Delta c = 0.0\%$) are much smaller than those for typical layered LiMO$_2$ family[21].

We calculate the Na hopping barrier by o-t-o pathway (moving from one octahedral site to an intermediate tetrahedral site, then to another octahedral site) in Na$_{0.5}$CrS$_2$ with 1/6 Cr/V′$_{Na}$ antisite. As shown in Supplementary Fig. 16, the calculated Na migration barrier of 0.33 eV is comparable to the typical 1-TM barrier values (~0.3 eV) in layered oxides[40], indicating the reduced

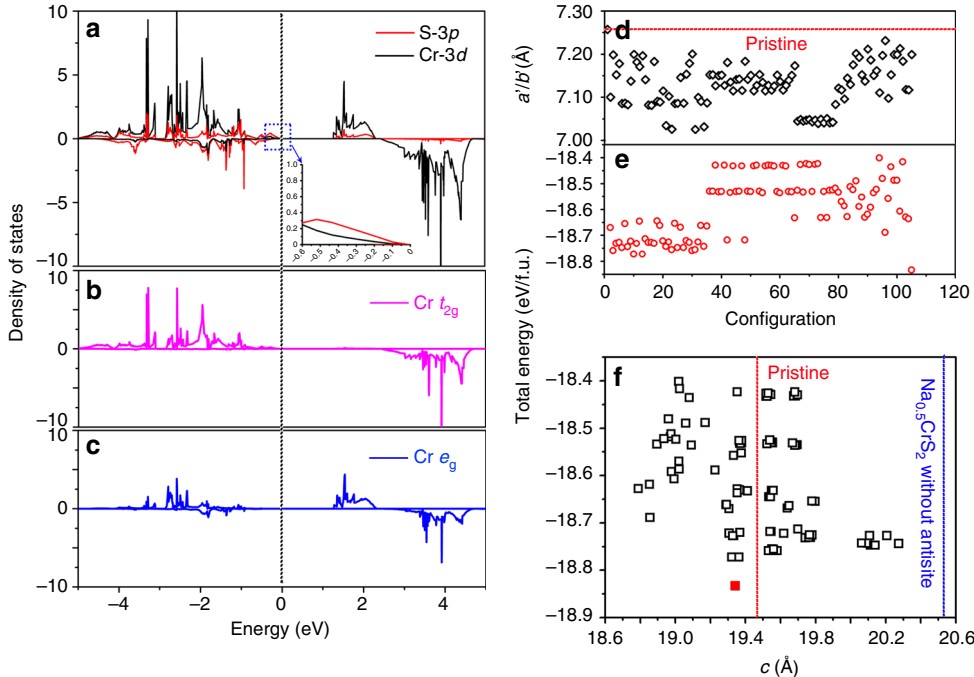

**Fig. 6** DOS and Cr/V′$_{Na}$ antisite. **a–c** Density of states of NaCrS$_2$. **d**, **e** Calculated lattice parameters $a'/b'$ and total energies of $2 \times 2 \times 1$ conventional cells and **f** dependence of calculated total energy of Na$_{0.5}$CrS$_2$ with 1/6 Cr/V′$_{Na}$ antisite on lattice parameter $c$. The total energy unit is eV/f.u.

tetrahedron height. This result suggests that the Cr/V′$_{Na}$ antisite has the positive effect on Na diffusion. As shown in Supplementary Fig. 17a, along with extraction of Na$^+$ from NaCrS$_2$, Cr first travels to the Na layer via the cross-layer migration, and then realizes the intra-layer migration, which acts as a "glue" between the Cr-S layers. It shortens the interlayer distance, so the length of $c$ axis decreases. Based on XRD, XAS, STEM, and first-principles calculation results, the "abnormal" unit cell breathing of NaCrS$_2$ upon charge is influenced significantly by cross-layer migration of Cr accompanied by an energy barrier of 0.83 eV (Supplementary Fig. 17b). It is noted that the energy barrier during the electrochemical charge/discharge process should be lower. Such a migration of Cr$^{3+}$ ions can reduce the lattice expansion along the $c$ direction, and then serves as 'pillar' to prevent the structure collapse.

In order to further characterize the electronic structure during the desodiation process, we plot the charge densities of pristine NaCrS$_2$ and Na$_{0.5}$CrS$_2$ and 1/6 Cr/V′$_{Na}$ antisite and summarize Cr-S, Na-S, and S-S distances related to all S ions in $2 \times 2 \times 2$ Na$_{0.5}$CrS$_2$ with 1/6 Cr/V′$_{Na}$ antisite conventional cell, as shown in Fig. 5d–e and Supplementary Table 5. As compared with NaCrS$_2$, the charge density distribution of Cr-3d states in Na$_{0.5}$CrS$_2$ is nearly unchanged, indicating that Cr ions do not take part in redox reaction during charge/discharge process. Instead, it is found that there are two pairs of dimers S$_2^{2-}$ with an S-S distance of 2.083 Å, two dangling S$^-$ ions with two-coordinated Cr ions and others are general S$^{2-}$ ions. It should be emphasized that the usage of the larger-size supercell is expected to give polysulfides ions. The previous XPS and XANES data shown in Figs. 4 and 5 can be well explained by the calculation results.

## Discussion

Previous studies on the electrochemistry of sulfur-based electrode materials mainly involve the conversion reaction between two or three species, such as between LiCrS$_2$ and Cr + Li$_2$S[7], between VS$_4$ and Li$_2$S + V[41], between CuCr$_2$S$_4$ and Cu + Cr$_2$S$_4$[15, 42], between

Li$_2$FeS$_2$ and FeS$_2$[43], and between sulfur and Li$_2$S[44]. The oxidation of S$^{2-}$ was revealed in some of layered sulfides. For example, the deintercalation of copper from Cu[Cr$_2$]S$_4$ with the placement of Cr$^{4+}$/Cr$^{3+}$ couples below the top of S-3p state results in the itinerant holes in the S-3p bands. However, copper could not be reversibly intercalated into [Cr$_2$]S$_4$[15]. Furthermore, the holes in the S-3p bands are not trapped out at S$_2^{2-}$ as discussed by Goodenough[42]. In layered sulfides of LiV$_{1-y}$M$_y$S$_2$ and LiTi$_{1-y}$M$_y$S$_2$ (M = Cr/Fe), the V(IV)/V(III) and Ti(IV)/Ti(III) couples are situated at the top of S-3p states. The holes in S-3p states could also be achieved during lithium removal. However, there are sufficient cation-3d characters from V or Ti enough to prevent the formation of p-p antibonding states. Dahn et al.[43] investigated the electrochemical mechanism of Li/FeS$_2$ and Li/Li$_2$FeS$_2$, their results indicated that the deintercalation of Li from Li$_2$FeS$_2$ involved Fe$^{3+}$S$^{2-}$(S$_2$)$^{2-}$$_{1/2}$, in which there is a strong Fe (3d)-S (3p) overlap[45, 46]. These works indicated the feasibility of inducing the anionic redox by forming the (S$_2$)$^{n-}$ species reversibly[47]. On the other hand, previously reported S-based intercalation-type compounds Li$_{1-x}$CrS$_2$ only delivered a low capacity of 30 mAh g$^{-1}$[42]. In this work, the reversible discharge capacity of 95 mAh g$^{-1}$ is presented. The very little change of lattice parameter $c$ and 100% coulombic efficiency during the cycles are quite impressive. By charging up to higher voltages, larger reversible capacities of 120 mAh g$^{-1}$ can be achieved (Supplementary Fig. 18), and a progressive cationic redox reaction may be involved. Our experimental and DFT calculation results demonstrate that the redox of sulfur is mainly triggered by unique band structure of S-3p of NaCrS$_2$ and isolated S-3p orbital introduced by Cr/V′$_{Na}$ antisite, and holes in sulfur orbital with a high concentration condense into dianion S$_2^{2-}$.

In layered compounds, a redox couple energy position relative to the top of an anion p band could determine the nature of deintercalation/intercalation reactions. The active TM$^n$/TM$^{n+1}$ redox couple as a single cationic redox reaction in the layered transition metal compounds, such as LiCoO$_2$, locates at the top of O$^{2-}$ p band (Supplementary Fig. 19a). The cumulative

Ru $(Ir)^n$/Ru $(Ir)^{n+1}$ and $O^{2-}$/$O_2^{2-}$ redox couples of layered $Li_2MO_3$[9, 10] as cationic and anionic redox reactions exhibit a partial overlap at higher potential due to the strong covalent mixing, which can expand the half occupied couple to an itinerant band between Ru (Ir) $d$ and $O^{2-}$ $p$ bands (Supplementary Fig. 19b). As shown in Fig. 6a, the $Cr^{4+}$/$Cr^{3+}$ couple in layered $NaCrS_2$ is pinned below the $S^{2-}$/$S_2^{2-}$ redox couple, so the empty antibonding states of the redox couple are dominated by S $p$ band leading to a single anionic redox reaction. The $S_2^{2-}$/$S^{2-}$ couple of $NaCrS_2$ provides a charge potential of 2.8 V for $Na^+$ deintercalation from the octahedral sites if the top of $S_2^{2-}$/$S^{2-}$ couple and the pinned Cr 3$d$ band are both stabilized by Cr migration into partially occupied Na layer. DFT calculation results indicate that the formation of $Cr/V'_{Na}$ antisite involves three substeps (Supplementary Fig. 17): Cr first travels from Cr site to Na layer via cross-layer migration, and then travels from Na vacancy to another by the distance of 3.446 Å. Finally, via divacancy mechanism, Na migrates from the original lattice site to adjacent vacancy by the distance of 3.629 Å. It means, Cr migration from Na-S-Cr configuration to Na vacancy changes the configuration symmetry around S, resulting in the formation of Na-S-□ (□: Vacancy) configuration. Thus, there will be an isolated S 3$p$ orbital, indicating the non-bonding sulfur hole states, which have ionic character rather than covalent character. At a larger concentration, these sulfur holes condense into dianion $S_2^{2-}$ as shown in Fig. 5e. The reaction potential based on the single anionic redox chemistry of $S^{2-}$/$S_2^{2-}$ is relatively low. Nevertheless, its unique features with "abnormal" unit cell breathing behavior (constant $c$ axis during charge and discharge) and redox chemistry of $S^{2-}$/$S_2^{2-}$ triggered by $Cr/V'_{Na}$ antisite enrich the in-depth understanding for the nature of redox reaction of layered intercalation compounds. The single anionic redox chemistry of $S^{2-}$/$S_2^{2-}$ may open a new research domain and provide new perspectives on how to design the composition and structure of high-capacity intercalation-type layered metal sulfides for rechargeable batteries by tuning the hybridization of other transition-metal $d$ and S 3$p$ bands or by designing double anion systems with O and S elements.

## Methods

**Sample preparation**. To synthesize the $NaCrS_2$ powder materials, a well-grounded mixture of $Na_2S$, S, and Cr in stoichiometry was placed into carbon-coated quartz tubes. They were heated to 900 °C, kept at that temperature for 6 h, and then cooled down slowly for over 3 h to 300 °C, followed by quenching. All preparation was performed under argon unless otherwise noted.

**Electrochemical characterization**. A slurry of $NaCrS_2$ (70 wt%), conductive carbon black (20 wt%), and polyvinylidenefluoride (Sigma-Aldrich, 10 wt%) dispersed in N-methyl-2-pyrrolidone (Sigma-Aldrich) was coated on aluminum foil. Two thousand thirty-two coin cells were used for electrochemical test. The electrolyte consisted of 1 M $NaClO_4$ dissolved in 1:1 (volume) ethylene carbonate/dimethyl carbonate. Electrochemical performance measurements were carried out on a LAND battery tester.

**In situ XRD measurements**. XRD data were carried out at National Synchrotron Light Source (beamline ×14A) at Brookhaven National Laboratory. The wavelength of the X-ray was 0.7747 Å. A home-made electrochemical cell with X-ray windows was used. The angles of XRD spectra were switched to the angles for Cu-K$a$ ($\lambda = 1.54$ Å), in order to compare with the literature easily.

**XAS measurements**. Ex situ Cr K-edge XAS spectra were measured at Advanced Photon Source (beamline 12BM) at Argonne National Laboratory. Ex situ S K-edge XAS spectra were obtained at beamline 4B7A in Beijing Synchrotron Radiation Facility. The bending magnet beamline covers the spectral range from 2050 to 5700 eV, with energy resolving power up to 7000 and a beam size of 3mm × 1 mm. The EXAFS and XANES spectra were processed using Artemis and Athena software packages[48].

**STEM measurements**. The detailed measurement set-up was described elsewhere[49].

**XPS characterization**. XPS was carried out on a PHI 5000C ESCA System with monochromatic Al-K$a$ X-ray source. The C 1s peak at 285.0 eV from hydrocarbon contamination was used to calibrate the binding energy.

**EPR characterization**. EPR spectra were collected on a Bruker EMX-8/2.7 spectrometer. Microwave power was set to 2 mV.

**DFT calculations**. Based on the projector-augmented wave method within DFT theory[50], conducted with the VASP program[51], ferromagnetic spin-polarized calculations were carried out. We used the Perdew-Burke-Ernzerhof functional for exchange correlation[52]. We set an effective $U_{eff}$ value to 3.5 eV as discussed in electronic structure calculations on $MCrS_2$ (M = Li, Na, K, and Ag)[53]. The plane wave cutoff energy and Monkhorst-Pack $k$-point mesh were set to 550 eV and $2 \times 2 \times 1$ for $NaCrS_2$ conventional cell, respectively. As for the calculation of the electronic density of states, $4 \times 4 \times 1$ $k$-point mesh for the conventional cell and the modified tetrahedron method were used. The above parameters made the total energy converged to 2 meV per atom. The calculated structural parameters of $NaCrS_2$ are consistent with experimental ones ($a$ and $c$: 3.627582 and 19.446446 Å vs. 3.5270 and 19.3506 Å). To determine the energy barriers for Na or Cr ion diffusion in $NaCrS_2$, the climbing-image nudged elastic band method[54] was employed for searching the minimum-energy path.

**Data availability**. The main data supporting the findings of this study are available within the article and its Supplementary Information files. Extra data are available from the corresponding author upon request.

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

## Acknowledgements

This work was supported by National Natural Science Foundation of China (U1430104, 51622207, 51502039, U1630134, and 51372228), National Key Research and Development Program of China (2016YFB0901504 and 2017YFB0701600), and Science & Technology Commission of Shanghai Municipality (11JC 1400500), and Shanghai Pujiang Program (14PJ1403900). The work at Brookhaven National Laboratory was supported by the Assistant Secretary for Energy Efficiency and Renewable Energy, Office of Vehicle Technologies of the U.S. Department of Energy through the Advanced Battery Materials Research (BMR) Program under Contract No. DE-SC0012704. The authors gratefully acknowledge the help by beamline scientists at 12BM of Advanced Photon Source at Argonne National Laboratory, supported by the U.S. Department of Energy under Contract No. DE-AC02-06CH11357. The authors also thank BL14W1 of Shanghai Synchrotron Radiation Facility, the high performance computing platform of Shanghai University, and national supercomputer center in Guangzhou.

## Author contributions

Z.-W.F. and S.-Q.S. supervised the research. Z.S., S.-Q.S., and Z.-W.F. wrote the manuscript. Z.S. and J.-L.Y. tested the electrochemical performance. Y.-N.Z. and X.-Q.Y. performed the XRD and XAS measurements. Z.S., J.-L.Y., L.-L.C., Q.W., and S.-Q.S. conducted the DFT calculations. N.Z. and X.L. performed the S K-edge XANES experiments and processed the data. L.G. conducted STEM imaging.

## Additional information

**Competing interests:** The authors declare no competing financial interests.

