## [Peer Review File · Nature Communications]

Reviewers' comments:

Reviewer #1 (Remarks to the Author):

Utilization of anion redox for charge compensation is a good strategy to extend the charge storage capability in battery systems. The paper reports a sole anion redox of S²⁻/S²²⁻ in NaCrS₂, which is an interesting topic and shows some interesting results. However, the authors never mention many previous important works. For example, Goodenough already reports a single O redox in Na_{0.6}(Li_{0.2}Mn_{0.8})O₂ in Energy and Environmental Science, which has a higher working voltage than S materials. The authors should evaluate the novelty of their work properly in the introduction. In addition, some questions need to be addressed as listed below. I would not recommend the publication of paper in current form and the authors need a major revision before the consideration for publication.

1. The authors make discussion on anion redox, especially oxygen redox in the introduction. The viewpoints are somehow not consistent with the demonstrations in those publications. The authors need to re-write this part.

2. The paper studies anion redox of S. The discussion about previous works on S materials is somewhat lacking. Some important works have been reported by Goodenough and Dahn. Failing to discuss fully how work here compares with done previously.

3. It would be good if the x values in Na_xCrS₂ would be added to electrochemistry in addition to capacity scale. Figure 1d and 1e are not labelled properly. Why do the authors show the charge capacity only in cycling curves?

4. I don't quite understand why the lattice parameter change of NaCrS₂ in a Na cell is different from that in a Li cell, given that the same material has been used. Also, the explanation on the basis of unit cell breathing is a bit unconvincing. Could the migration of Cr³⁺ to Na layer lead to shrinkage along c direction, considering their size difference? More reasonable explanation needs to be organized for the structural evolution in this process.

5. What does the XRD look like in the subsequent cycles? Does the cation migration lead to the formation of rock salt structure? Can you quantify the cation mismatch using STEM?

6. Why do the authors collect EPR data at 100K? Charge transfer may occur at low temperature. If EPR does not provide any useful information on oxidation of S, I would put it in SI.

Reviewer #2 (Remarks to the Author):

In this manuscript, the author describes the electrochemical behavior of the Na_xCrS₂ phase upon sodium intercalation/deintercalation. The authors employ a battery of characterization to demonstrate that Cr migration in antisite triggers the redox activity of sulfur in the phase. I had pleasure reading this paper, and I feel that it deserves publication. The very unique demonstration of cationic migration triggering the anionic redox process is very beautiful. Nevertheless, prior to publication, few issues must be fixed.

In details.

In the introduction :

The anionic redox is not triggered by lowering the cationic band closer to the oxygen band, but it is triggered by forming non-bonding oxygen states that are created depending on the symmetry when O(p) orbitals don't overlap with M(3d) orbital. Hence, the increased capacity arises from the fact that two redox bands can be accessed (MO* states and non-bonding oxygen states), rather than by increasing the covalent character of the M-O bond as written in the introduction.

This part should be re-written.

In the first part describing CV results, the author shouldn't describe yet these two peaks as cationic

and anionic, as the demonstration is made only latter.

There are some issues with the numbering of the Figures in the text referring to figures in the SI, that should be fixed.

I feel that the most interesting part and the main beauty of this work is the demonstration that cationic migration can eventually trigger the anionic redox, and so that an activation step is necessary so to enable the full potential of the material. My feeling is that the STEM results are critical for that, especially Figure S5 when compared to S2. So, I feel that these figures should be in the main text, rather than in the SI since they bring the demonstration that cationic migration is indeed happening during cycling.

Concerning the EPR results, the fact that no S-signal is detected is well explained by the formation of $(S_2)^{2-}$ species that have no unpaired electron and are therefore inactive. So it would mean that the redox involved is between S^{2-} and $(S_2)^{2-}$. This type of redox process corresponds to what has been proposed for layered sulfide compounds by Rouxel. Therefore, I don't understand why the authors explain that the fact that no S signal can be detected is coming from instability. The sulfur redox indeed induces the formation of different species than the oxygen redox for which $(O_2)^{3-}$ species are formed, the author should simply interpret their results simply describing this redox and not trying to compare it with results from oxide compounds for LIBs.

Concerning the XAS at the S K-edge, the involvement of sulfur redox is different from a change in the pre-edge features increase or decrease that would correspond simply to a more covalent bond (i.e. implication of MS^* bonds rather than pure Sp states). The formation of $(S_2)^{2-}$ should lead to the formation of a new peak, different from those corresponding to t_{2g} and e_g states.

They are numerous typos and grammatical errors throughout the text, they should be fixed.

REVIEWERS' COMMENTS:

Reviewer #1 (Remarks to the Author):

The authors revised the manuscript very carefully. There are still some typos in the manuscript, but I am happy to recommend it for publication.

Reviewer #2 (Remarks to the Author):

The authors have responded to the questions asked by the reviewer. The manuscript can now be accepted.

Thank you very much for your kind consideration of manuscript entitled “**Antisite occupation induced single anionic redox chemistry and structural stabilization of layered NaCrS₂**” (Ms. No.: NCOMMS-17-07192). We sincerely appreciate the valuable and in-depth suggestions and comments made by the reviewers. Their suggestions made great improvements for this manuscript. Our manuscript has been revised carefully according to the reviewer’s comments. We have addressed all the concerns of reviewers and the revisions are highlighted in text. Some new experiments have been done and the results are helpful to clarify the key concern from the reviewers. The changes and responses to the reviewers are explained below sentence by sentence:

To Reviewer #1

Comments:

Utilization of anion redox for charge compensation is a good strategy to extend the charge storage capability in battery systems. The paper reports a sole anion redox of S²⁻/S₂²⁻ in NaCrS₂, which is an interesting topic and shows some interesting results. However, the authors never mention many **previous important works**. For example, Goodenough already reports a single O redox in Na_{0.6}(Li_{0.2}Mn_{0.8})O₂ in *Energy and Environmental Science*, which has a higher working voltage than S materials. **The authors should evaluate the novelty of their work properly in the introduction.** In addition, some questions need to be addressed as listed below. I would not recommend the publication of paper in current form and the authors need a major revision before the consideration for publication.

Reply:

Thanks for reviewer’s encouragement and suggestion.

Two earlier publications about the sole anion redox in layered compounds were found and added in the revised manuscript. According to the reviewer’s suggestion, the previous work about Na_{0.6}(Li_{0.2}Mn_{0.8})O₂ cathode in NIB reported by Goodenough (*Energy Environ. Sci.* 9, 2575-2577 (2016)) has been cited and discussed in the introduction section. Another work about a S-based intercalation-type compounds Li_{1-x}CrS₂ with a low discharge capacity of 30 mAh/g⁴², (*J. Solid State Chem.* 182, 2904-2911 (2009)) can be found in the discussion part of the original manuscript.

The sentences “Recently, Du and Goodenough et al. investigated the sole anionic redox in a P3-layered Na_{0.6}(Li_{0.2}Mn_{0.8})O₂, in which the Mn⁺⁵/Mn⁺⁴ redox energy lies far below the top of the O-2p bands. They found that the holes into O-2p bands were introduced along with desodiation, but cannot cycle reversibly¹⁶.” have been added in the introduction of the revised manuscript in Line 15, Page 4.

Comment 1:

The authors make discussion on anion redox, especially oxygen redox in the introduction. The viewpoints are somehow not consistent with the demonstrations in those publications. The authors need to re-write this part.

Reply:

The reviewer is right. The discussions on oxygen redox in the introduction are somehow not consistent with the demonstrations in those publications.

According to reviewer's suggestion, we have re-written these discussions about origin of oxygen redox in Li rich layered material in the introduction section as follows:

“Recent studies revealed that the oxygen redox was the result of a lowered transition-metal d-band with respect to the oxygen 2p band. In this case, the stronger overlap of metal d and oxygen p-band triggers the anionic redox activity and contributes the extra capacity. The degree of cation-d-anion-p band mixing is intrinsically determined by the electronegativity of metal and ligand ions.” in the original manuscript has been replaced with “recent studies revealed that **that the anionic redox was triggered by forming non-bonding oxygen states, which was created due to that those O 2p orbitals along the Li-O-Li configurations have no metal orbitals with which it can hybridize, nor does it hybridize with the lithium Li 2s orbital. The oxygen oxidation occurs from the orphaned Li-O-Li states and the increased capacity arises from the fact that two redox bands can be access to MO* states and non-bonding oxygen states in Li-excess layered or cation disordered materials**” in the introduction of the revised manuscript.

Comment 2:

The paper studies anion redox of S. The discussion about previous works on S materials is somewhat lacking. Some important works have been reported by Goodenough and Dahn. Failing discuss fully how work here compares with done previously.

Reply:

According to the reviewer's suggestion, the previous works about S materials reported by Goodenough, J.Rouxel and J.R. Dahn were cited and discussed in the revised manuscript.

These discussions have been added in the revised manuscript in Line 11, Page 17 as follows:

The oxidation of S²⁻ in some layered sulfides was revealed. For example, the deintercalation of copper from Cu[Cr₂]S₄ with the placement of Cr(IV)/Cr(III) couples below the top of the S-3p bands result in the itinerant holes in the S-3p bands. However, copper could not be reversibly intercalated into Cu[Cr₂]S₄¹⁵. Furthermore, the holes in the S-3p bands are not trapped out at S₂²⁻ as discussed by Goodenough⁴². In layered sulfides of LiV_{1-y}M_yS₂ and LiTi_{1-y}M_yS₂ (M=Cr/Fe), the V(IV)/V(III) and Ti(IV)/Ti(III) couples are pinned at the top of S-3p bands. The holes in S-3p band could also be achieved during lithium removal. However, there are sufficient

cation-3d characters from V or Ti enough to prevent the formation of *p-p* antibonding states. Dahn⁴³ investigated the electrochemical mechanism of Li/FeS₂ and Li/Li₂FeS₂, their results indicated that the deintercalation of Li from Li₂FeS₂ involved an oxidation of Fe²⁺ to Fe³⁺ followed by the formation of Fe³⁺S²⁻(S₂)^{2-_{1/2}}, in which there is a strong Fe (3d)-S (3p) overlap⁴⁵⁻⁴⁶. These works indicated the feasibility of triggering anionic redox processes through reversible formation of (S₂)ⁿ⁻ species⁴⁷. Our experimental and DFT calculation results have demonstrated that the redox of sulfur is mainly triggered by unique band structure of S-3p and isolated S 3p orbital introduced by Cr/V_{Na} antisite, and holes in sulfur orbital with a high concentration condense into dianion S₂²⁻.

Comment 3:

It would be good the *x* values in Na_{*x*}CrS₂ would be added to electrochemistry in addition to capacity scale. Figure 1d and 1e are not labelled properly. Why do the authors show the charge capacity only in cycling curves?

Reply:

Thanks for reviewer's suggestion.

The *x* values in Na_{*x*}CrS₂ have been added in Fig. 1c and Supplementary Fig. 4a.

The labels in Fig. 1d and 1e have been corrected.

The discharge capacities in cycling curves have been added in Fig.1d and Supplementary Fig. 4b.

Comment 4:

I don't quite understand why the lattice parameter change of NaCrS₂ in a Na cell is different from that in a Li cell, given that the same material has been used. Also, the explanation on the basis of unit cell breathing is a bit unconvincing. Could the migration of Cr³⁺ to Na layer lead to shrinkage along *c* direction, considering their size difference? More reasonable explanation needs to be organized for the structural evolution in this process.

Reply:

Thanks for reviewer's suggestion.

As the reviewer pointed out, the lattice parameter change of NaCrS₂ electrode during the first charging process in Na cell should be similar to that in the Li cell.

Figure R1. *In situ* XRD pattern of NaCrS₂ electrode during the first charging process in Li cell.

In order to compare the structure evolution of NaCrS₂ electrode in Li and Na cells, the *in situ* XRD of NaCrS₂ electrode during the first charging process in Li cell was carried out and the results were added in Supplementary Fig. 7. As shown in Figure R1, the (003) diffraction peak almost keeps unchanged, while the (110) peak gradually moves towards higher 2θ angle with the extraction of Na from NaCrS₂. It means that, the lattice parameter c keeps unchanged and lattice parameter a/b decreases during the charging process. Obviously, these results are similar with the lattice change of NaCrS₂ electrode in Na cell. We noted that the structure evolution of NaCrS₂ in lithium battery here is different from the previous report [A. GUSHEV et. al. Solid State Ionics 13 (1984) 181-190], where the lattice parameter c increases from 19.51 Å for NaCrS₂ to 20.19 Å for Na_{0.5}CrS₂ while the lattice parameter a is almost unchanged during the deintercalation. The difference of the lattice parameter changes of NaCrS₂ may be attributed to the different synthesis methods and experimental conditions such as precursor, synthesis temperature and reaction time. Another possibility is that they obtained the lattice parameters from ex situ XRD experiment. The sample could be change during the ex situ sample preparation especially for the charged one.

To our knowledge, the size differences are not key factors in the migration of Cr³⁺ to Na layer leading to shrinkage along c direction.

According to the reviewer' suggestion, we re-organize these reasonable explanations for the structural evolution as follows:

Because the layered structure of NaCrS₂ with edge-sharing of CrO₆ octahedra is different from that of LiNbO₂ with edge-sharing of NbO₆ trigonal prisms, the

mechanism of unchanged parameter c in layered LiNbO_2 could not be used for that in NaCrS_2 .

Such Cr migration into Na sites may be responsible for the unchanged lattice parameter c .

Comment 5:

What does the XRD look like in the subsequent cycles? Does the cation migration lead to the formation of rock salt structure? Can you quantify the cation mismatch using STEM?

Figure R2 *In situ* XRD pattern of NaCrS_2 electrode during the 3rd charge-discharge process in Na cell.

Reply:

The *in situ* XRD patterns for NaCrS_2 electrode in a Na cell during the 3rd cycle have been added in Supplementary Fig. 6 according to the reviewer's suggestion.

Regarding to reviewer's concern about rock salt phase formation, we did not find that the cation migration lead to the formation of rock salt structure. As shown in Figure R2, it can be seen that the structure evolution and lattice parameter change of NaCrS_2 electrode during the 3rd cycle are almost consistent with those during the 1st cycle. No any new peaks are observed, indicating that no new structure is formed in the subsequent cycles. We have added these discussions in the revised manuscript as follow:

The structure evolution and lattice parameter change of NaCrS_2 electrode during the 3rd cycle are found to be almost consistent with those during the 1st cycle (Supplementary Fig. 6). No any other peaks is observed, indicating that no new structure is formed in the subsequent cycles.

By carrying out multislice simulation of structures with different quantity of cation mismatch, the cation mismatch in $\text{Na}_{0.5}\text{CrS}_2$ can be quantified. However, the structure

of $\text{Na}_{0.5}\text{CrS}_2$ is very complicated, because of the uncertain distribution of Na, S and vacancies, which makes quantification of the mismatch very difficult. By extracting the intensity line profile of Na layer in HAADF-STEM image, we can clearly see the contrast different between NaCrS_2 and $\text{Na}_{0.5}\text{CrS}_2$. In Supplementary Fig. 1b, the intensity ratio between Na and Cr column is $\sim 14.59\%$. Meanwhile, this ratio in $\text{Na}_{0.5}\text{CrS}_2$ is $\sim 49.81\%$ as shown in Fig. 3b, which clearly show the cation mismatch in $\text{Na}_{0.5}\text{CrS}_2$. This phenomenon has been found in LiNiO_2 , LiCrO_2 and LiCoO_2 cathodes. Furthermore, the fluctuation intensity in Na layer was observed clearly from Fig. 3c, which means uneven occupation of Cr in Na layer. This confirms the complexity and uneven of the distribution of Na vacancies. However, according to DFT calculation result, $\text{Cr}/\text{V}_{\text{Na}}$ antisite with ratio of 1/6 obtained in $\text{Na}_{0.5}\text{CrS}_2$ can be found in the original manuscript.

Comment 6:

Why do the authors collect EPR data at 100K? Charge transfer may occur at low temperature. If EPR does not provide any useful information on oxidation of S, I would put it in SI.

Reply:

We have deleted EPR data at 100K according to the reviewer's suggestion.

EPR data at the RT can not only confirm that the valance of Cr ions keeps unchanged, but also provide an information on the possible formation of $(\text{S}_2)^{2-}$ species, which have no unpaired electron and are EPR-inactive. Thus, we would like to keep it in the revised manuscript. The useful information on oxidation of S in EPR S-signal has been added in the revised manuscript as follow:

In addition, no S-signal is detected in the charged sample, which can be explained by the formation of $(\text{S}_2)^{2-}$ species that have no unpaired electron as shown in Supplementary Fig. 8, and are therefore EPR-inactive.

To Reviewer #2

In this manuscript, the author describes the electrochemical behavior of the Na_xCrS_2 phase upon sodium intercalation/deintercalation. The authors employ a battery of characterization to demonstrate that Cr migration in antisite triggers the redox activity of sulfur in the phase. I had pleasure reading this paper, and I feel that it deserves publication. The very unique demonstration of cationic migration triggering the anionic redox process is very beautiful. Nevertheless, prior to publication, few issues must be fixed.

In details.

Reply

Thanks for reviewer's encouragement.

Comment:

In the introduction:

The anionic redox is not triggered by lowering the cationic band closer to the oxygen band, but it is triggered by forming non-bonding oxygen states that are created depending on the symmetry when O (p) orbitals don't overlap with M ($3d$) orbital. Hence, the increased capacity arises from the fact that two redox bands can be access (MO^* states and non-bonding oxygen states), rather than by increasing the covalent character of the M-O bond as written in the introduction.

This part should be re-written.

Reply:

The reviewer is right. As reviewer pointed out, the anionic redox was triggered by forming non-bonding anion hole states created depending on the configuration symmetry when O (p) orbitals don't overlap with M ($3d$) orbital. We have re-written this part in the revised manuscript in Line 5 Page 4 as follow:

recent studies revealed that the anionic redox was triggered by forming non-bonding oxygen states, which was created due to that those O $2p$ orbitals along the Li-O-Li configurations have no metal orbitals with which it can hybridize, nor does it hybridize with the lithium Li $2s$ orbital. The oxygen oxidation occurs from the orphaned Li-O-Li states and the increased capacity arises from the fact that two redox bands can be access by MO^* states and non-bonding oxygen states in Li-excess layered or cation disordered materials.

Comment:

In the first part describing CV results, the author shouldn't describe yet these two peaks as cationic and anionic, as the demonstration is made only latter.

Reply:

According to the reviewer's suggestion, the related description of CV results about these two cationic and anionic peaks have been deleted.

Comment:

There are some issues with the numbering of the Figures in the text referring to figures in the SI that should be fixed.

Reply:

Thank the reviewer's suggestion. The number of the figures in the main text was corrected in the revised manuscript.

Comment:

I feel that the most interesting part and the main beauty of this work is the demonstration that cationic migration can eventually trigger the anionic redox, and so that an activation step is necessary so to enable the full potential of the material. My feeling is that the STEM results are critical for that, especially Figure S5 when compared to S2. So, I feel that these figures should be in the main text, rather than in the SI since they bring the demonstration that cationic migration is indeed happening during cycling.

Reply:

Thanks for reviewer's suggestion. As reviewer pointed out, the cation migration changes the configuration symmetry around S, resulting in the formation of non-bonding sulfur hole states, thus can trigger the sulfur redox.

According to the reviewer's suggestion, the STEM results of pristine and charged samples have been placed in Fig. 1b and Fig. 3 in the revised manuscript, respectively.

Comments:

Concerning the EPR results, the fact that no S^- signal is detected is well explained by the formation of $(S_2)^{2-}$ species that have no unpaired electron and are therefore inactive. So it would mean that the redox involved is between S^{2-} and $(S_2)^{2-}$. This type of redox process corresponds to what has been proposed for layered sulfide compounds by Rouxel. Therefore, I don't understand why the authors explain that the fact that no S signal can be detected is coming from instability. The sulfur redox indeed induces the formation of different species than the oxygen redox for which $(O_2)^{3-}$ species are formed, the author should simply interpret their results simply describing this redox and not trying to compare it with results from oxide compounds for LIBs.

Reply:

Thanks for reviewer's suggestion. As reviewer pointed out, no S^- signal is detected is well explained by the formation of $(S_2)^{2-}$ species that have no unpaired electron and are therefore inactive.

According to reviewer's suggestion, the description of S-EPR spectrum has been corrected in the revised manuscript as follows:

In addition, no S-signal is detected in the charged sample, which can be explained by the formation of $(S_2)^{2-}$ species that have no unpaired electron as shown in Supplementary Fig. 8, and are therefore EPR-inactive.

Comments:

Concerning the XAS at the S K-edge, the involvement of sulfur redox is different from a change in the pre-edge features increase or decrease that would correspond simply to a more covalent bond (i.e. implication of MS^* bonds rather than pure S p states). The formation of $(S_2)^{2-}$ should lead to the formation of a new peak, different from those corresponding to t_{2g} and e_g states.

Reply:

Thanks for reviewer's suggestion. As reviewer pointed out, formation of $(S_2)^{2-}$ should lead to the formation of a new peak at the higher energy position.

The position shifts of the edge peak A and the intensity increments of the edge peak B during the charge should mainly result from the contributions of the formation of new chemical bonds to the sulfur atoms. This new peak might emerge around 2742 eV with increasing intensity during the charge process. However, this new peak is superposed on the peak B in the pristine, the spectral resolution is not high enough to experimentally separate the contributions from different bonds. In order to avoid the confusion, we have added some sentences in the revised manuscript as follow:

The position shifts of the edge peak A and the intensity increments of the edge peak B during the charge should mainly result from the contributions of the formation of new chemical bonds to the sulfur atoms. This new peak might emerge around 2742.1 eV and its intensity increases during the charge process. Therefore, such a new peak indicates the oxidation of S^{2-} during the sodium deintercalation process.

Comments:

They are numerous typos and grammatical errors throughout the text, they should be fixed.

Reply:

Thanks for reviewer's kind suggestion. The grammatical errors have been carefully corrected in the revised manuscript.

All the revised parts are highlighted with yellow background. If you need further information, please let me know. I am looking forward to hearing from you.